# Backhaul Capacity-Limited Joint User Association and Power Allocation Scheme in Ultra-Dense Millimeter-Wave Networks

**DOI:** 10.3390/e25030409

**Published:** 2023-02-23

**Authors:** Zhiwei Si, Gang Chuai, Kaisa Zhang, Weidong Gao, Xiangyu Chen, Xuewen Liu

**Affiliations:** 1Key Laboratory of Universal Wireless Communications, Ministry of Education, Beijing University of Posts and Telecommunications, Beijing 100876, China; 2School of Electronic Engineering, Beijing University of Posts and Telecommunications, Beijing 100876, China; 3School of Electronics and Communication Engineering, Beijing Electronics Science and Technology Institute, Beijing 100070, China

**Keywords:** millimeter-wave communication, coordinated multipoint transmission, ultra-dense network, many-to-many matching, successive convex approximation

## Abstract

Millimeter-wave (mmWave) communication is considered a promising technology for fifth-generation (5G) wireless communications systems since it can greatly improve system throughput. Unfortunately, because of extremely high frequency, mmWave transmission suffers from the signal blocking problem, which leads to the deterioration of transmission performance. In this paper, we solve this problem by the combination of ultra-dense network (UDN) and user-centric virtual cell architecture. The deployment of dense small base stations (SBSs) in UDN can reduce transmission distance of signals. The user-centric virtual cell architecture mitigates and exploits interference to improve throughput by using coordinated multipoint (CoMP) transmission technology. Nonetheless, the backhaul burden is heavy and interbeam interference still severe. Therefore, we propose a novel iterative backhaul capacity-limited joint user association and power allocation (JUAPA) scheme in ultra-dense mmWave networks under user-centric virtual cell architecture. To mitigate interference and satisfy quality of service (QoS) requirements of users, a nonconvex system throughput optimization problem is formulated. To solve this intractable optimization problem, we divide it into two alternating optimization subproblems, i.e., user association and power allocation. During each iteration, a many-to-many matching algorithm is designed to solve user association. Subsequently, we perform power allocation optimization using a successive convex approximation (SCA) algorithm. The results confirm that the performance of the proposed scheme is close to that of the exhaustive searching scheme, which greatly reduces complexity, and clearly superior to that of traditional schemes in improving system throughput and satisfying QoS requirements.

## 1. Introduction

Recently, the communication demand for 5G communication systems has exploded with the significant proliferation of smart devices and various advanced applications [1,2]. To meet communication demands, there is growing interest in millimeter-wave (mmWave) communications [3,4]. MmWave communication is considered to be a promising technology in 5G and beyond 5G communication systems [5]. Unfortunately, because of extremely high frequency, the performance of mmWave signal is severely affected by signal blockages [6]. This means that blockages, such as forests and buildings, make wireless channels between the transmitter and receiver non-line-of-sight (NLoS), resulting in a rapid degradation of the received signal quality. Therefore, it is critical to maintain the connection in line-of-sight (LoS) to mitigate blockage effects in mmWave communications.

In this paper, to solve the signal blockages problem, we combine mmWave communication with ultra-dense networks (UDNs) and virtual cell structures. By densely deploying multiple small base stations (SBSs), UDN can greatly increase the transmission rate [7,8]. Dense deployment of mmWave SBSs facilitates communications with short-range LoS propagation. However, ultra-dense mmWave communication networks bring severe interbeam interference [9]. To fully exploit the potential of UDN, system network architecture is considered to change from traditional cell-centric to user-centric [10]. User-centric virtual cell architecture mitigates and exploits interference to improve throughput by using coordinated multipoint (CoMP) transmission technology [11,12,13]. Ultra-dense mmWave networks and CoMP can increase system performance, which can bring some new challenges, such as heavy backhaul burden [14,15] and unavoidable interbeam interference [16]. To overcome these challenges, the third-generation partnership project (3GPP) proposed an integrated access and backhaul (IAB) architecture for 5G networks, where the same infrastructure and spectrum resources are used for access and backhaul [17]. The system capacity with infinite backhaul is easy to calculate. However, when considering finite backhaul, the information theory analysis of multiple SBSs becomes significantly complicated [18]. Limited backhaul capacity may cause traffic overload in SBSs with good access link quality [19]. Therefore, it is critical to propose a novel backhaul capacity-limited joint user association and power allocation scheme in user-centric ultra-dense mmWave communication networks.

In our previous work [20], we proposed a many-to-many matching algorithm to optimize user association and maximize system throughput with fixed power in ultra-dense mmWave communication networks, which reflects the substantial influence of different user association schemes on the system throughput. Ultra-dense mmWave communication will result in significant power consumption and heavy backhaul burden. However, we did not consider backhaul capacity constraints and power allocation optimization in our previous work. Therefore, as an extension of our previous paper [20], we investigate the backhaul capacity-limited joint user association and power allocation problem. Then, we describe the intercluster and intracluster interference among users and show its impact on user rate. Unlike previous studies, a joint user association and power allocation scheme is designed using alternating optimization, which effectively mitigates mmWave interbeam interference and improves system throughput and user rate. The main contributions of our paper are summarized as follows:A unified optimization framework is proposed for the system throughput maximization problem by jointly considering user association and power allocation. To effectively solve the intractable mixed-integer nonlinear programming optimization problem, we decouple it into two independent subproblems of user association and power allocation, which can be solved iteratively. The proposed algorithm has low computation complexity and can be extended to large-scale mmWave networks.We propose a novel many-to-many matching user association (MMUA) algorithm with externalities, which only has linear complexity. Then, its convergence and stability are verified theoretically. Subsequently, a low-complexity successive convex approximation (SCA) algorithm is designed to optimize power allocation, where Lagrangian dual decomposition is applied to solve the converted convex power allocation problem.The effectiveness and convergence of the proposed algorithm are verified by system simulations with different network parameter settings. The results confirm that the performance of the proposed algorithm is close to that of exhaustive searching algorithms and significantly superior to that of traditional algorithms in improving system throughput and satisfying QoS requirements, but with much reduced complexity.

The rest of our paper is organized as follows. Section 2 gives the related works. In Section 3, the system model and optimization problem formulation are presented in detail. Section 4 describes the proposed joint user association and power allocation optimization scheme and discusses its implementation process. Numerical results and discussion are given in Section 5. Finally, we conclude this paper in Section 6.

## 2. Related Works

There has been much research investigating user association and power allocation in mmWave communication networks. User association in mmWave communication networks was studied in [21]. In [22], the user association problem of maximizing spectrum and energy efficiency in mmWave backhaul networks was investigated. An iterative joint user association and power allocation algorithm for a single-band access scheme was designed by utilizing the Lagrange dual decomposition and Newton–Raphson method, and an approximate optimal solution for a multiband access scheme based on Markov approximation framework was obtained in [23]. In [24], authors proposed a coordinated user association and spectrum allocation by utilizing noncooperative game theory. Ref [25] proposed a user association algorithm with near-optimal performance and polynomial time computational complexity and performed joint power allocation and subchannel assignment using the difference of convex programming. A nondominated sorting genetic algorithm was designed in [26] to obtain a suboptimal solution of multiple associations which maximize energy efficiency while balancing base station load with different QoS values. In [27], based on multiagent reinforcement learning, the authors proposed a scalable and flexible algorithm for user association. In this method, as independent agents, users learned to autonomously coordinate their actions to maximize the network sum-rate. A multiobjective Harris hawk optimization algorithm was introduced in [28] to achieve near-optimal performance.

As the number of SBSs increases, the amount of data transmitted on the access link may exceed the upper limit of the data provided by the backhaul link, which degrades the system performance. There is some existing work on mitigating the effects of backhaul limitations, which can generally be classified into two categories: structured code-based approaches and data-compression-based approaches. Structured code-based approaches can improve system throughput under limited backhaul capacity by eliminate interference in wireless communication networks [29,30,31,32]. By using the decompression and channel decoding approach, ref. [33] proposed a per-base-station successive interference cancellation (SIC) scheme with limited backhual capacity, which maximized system throughput and significantly reduced complexity compared with the conventional joint decoding scheme. Data-compression-based approaches [34,35] reduced the amount of backhaul data through some compression or quantization techniques. In [36], an iterative algorithm based on the Majorization Minimization (MM) approach was proposed under the constraints of backhaul capacity, which guaranteed convergence to a stationary point of the sum-rate maximization problem for cloud radio access networks.

Recently, some studies have focused on reducing the computational complexity of user association and power allocation. With the advantages of low complexity, matching theory was widely used to solve the user association problem [37,38]. Ref. [39] designed a many-to-one matching user association algorithm. In [40], a many-to-one matching user association algorithm was given by considering mmWave backhaul links. Considering interference cancellation and backhaul capacity, ref. [41] proposed a two-step iterative scheme and iteratively solved it using many-to-many matching and SCA methods, respectively. In addition, to reduce computational complexity and maximize system throughput, ref. [42,43,44] utilized the difference of convex programming method to transform a nonconvex function to the difference of two convex functions and further obtain the suboptimal solution iteratively.

## 3. System Model and Problem Formulation

### 3.1. System Model

We consider the downlink mmWave transmission scenario of a heterogeneous network including a macro base station (MBS) and multiple small base stations (SBSs), which is shown in Figure 1. Assume user equipment (UEs) and mmWave SBSs are randomly distributed within the coverage area of MBS. The index of MBS is denoted by 0, and the index sets of mmWave SBSs and UEs are given by $N=\left\{1,2,\cdots ,N\right\}$ and $K=\left\{1,2,\cdots ,K\right\}$, respectively. In this network, access and backhaul (IAB) architecture [17] is adopted. The links between SBSs and MBS are backhaul links, and the links between UEs and SBSs are access links. It is assumed that all access and backhaul transmission links are operated in mmWave band.

### 3.2. Mmwave Signal Propagation Model

(1) Blockage Model: MmWave signals are susceptible to physical blockages in the network. Since the line-of-sight (LoS) and non-line-of-sight (NLoS ) pathloss characteristics are significantly different for mmWave signals, both LoS and NLoS path should be analyzed. The probability of each LoS path is pLoSdn,k=exp−dn,k−dn,kρρ, and the probability of each NLoS path is $p_{NLoS}\left(d_{n,k}\right)=1-p_{LoS}\left(d_{n,k}\right)$, where $d_{n,k}$ is the distance from transmitter *n* to receiver *k*, and ρ is LoS range constant.

(2) Effective Antenna Gain: The beamforming-based directional communication [45] is applied to calculate beam gain, where the beam gain of the directional beam is constant in mainlobe, and the beam gain in sidelobe is a small constant 0<ε≪1. Denote $\theta_{n}^{t}$ and $\theta_{n}^{r}$ as beamwidths of transmitter and receiver. $\varphi_{nk}^{t}$ and $\varphi_{kn}^{r}$ represent the beam boresight angles’ transmitter and receiver, respectively. $\zeta_{nk}^{t}$ and $\zeta_{kn}^{r}$ are the geographical angles’ transmitter and receiver. Therefore, transmission beam gain at mmWave SBS *n* is expressed as
(1)$$g_{nk,kn}^{t}\left(\theta_{n}^{t},\varphi_{nk}^{t},\zeta_{nk}^{t}\right)=\left\{\begin{array}{c} \varepsilon ,\;\;if\;\frac{\theta_{n}^{t}}{2}<\left|\varphi_{nk}^{t}-\zeta_{nk}^{t}\right|<2\pi -\frac{\theta_{n}^{t}}{2}\\ \frac{2\pi -\left(2\pi -\theta_{n}^{t}\right)\varepsilon }{\theta_{n}^{t}},\;\;\;\mathrm{otherwise}. \end{array}\right.$$

Similarly, the reception beam gain at UE *k* is calculated as
(2)$$g_{kn,nk}^{r}\left(\theta_{k}^{r},\varphi_{kn}^{r},\zeta_{kn}^{r}\right)=\left\{\begin{array}{c} \varepsilon ,\;\;if\;\frac{\theta_{k}^{r}}{2}<\left|\varphi_{kn}^{r}-\zeta_{kn}^{r}\right|<2\pi -\frac{\theta_{k}^{r}}{2}\\ \frac{2\pi -\left(2\pi -\theta_{k}^{r}\right)\varepsilon }{\theta_{k}^{r}},\;\;\;\mathrm{otherwise}. \end{array}\right.$$

Therefore, we define $G_{n,k}$ as the total beam gain between *n* and *k*, which is given by $G_{n,k}=g_{nk,kn}^{t}g_{kn,nk}^{r},\forall k\in K,n\in N$. Similarly, the total beam gain $G_{0,n}$ between MBS 0 and SBS *n* can be obtained in this way.

Let matrix *X* be the user association indicator matrix, where the variable $x_{k,n}\in \left\{0,1\right\},\forall n\in N,k\in K$, which can be given as
(3)$$\begin{array}{c} x_{k,n}=\left\{\begin{array}{cc} 1, & if\;UE_{k}\;\mathrm{is}\;\mathrm{associated}\;\mathrm{with}\;\mathrm{SB}S_{n},\\ 0, & \mathrm{otherwise}.\;\;\;\;\;\;\;\;\;\;\;\;\;\;\;\;\;\;\;\;\;\;\;\;\;\;\; \end{array}\right. \end{array}$$

Define $C_{k}\left(C_{k}\subseteq N\right)$ as the coordinated cluster for UE *k*, which consists of the SBSs satisfying $x_{k,n}=1$. Matrix *P* is defined as the power allocation matrix, where $p_{n,k},\forall k\in K,$n∈N, and $p_{n,k}$ is the transmit power from mmWave SBS *n* to UE *k*. UEs will suffer severe directive beam interference. Denote $I_{k,intra}$ and $I_{k,inter}$ as intracluster and intercluster interference power, which are given by (4) and (5), respectively.
(4)$$I_{k,intra}=\sum_{n\in C_{k}}\sum_{k^{\prime }\in K\setminus k}p_{n,k^{\prime }}{\bar{G}}_{nk^{\prime },kn}h_{n,k}$$
(5)$$I_{k,inter}=\sum_{n^{\prime }\in N\setminus C_{k}}\sum_{k^{\prime }\in K\setminus k}p_{n^{\prime },k^{\prime }}{\bar{G}}_{n^{\prime }k^{\prime },kn}h_{n^{\prime },k}$$
where ${\bar{G}}_{nk^{\prime },kn}=g_{nk^{\prime },kn}^{t}g_{kn,nk^{\prime }}^{r}$, ${\bar{G}}_{n^{\prime }k^{\prime },kn}=g_{n^{\prime }k^{\prime },kn}^{t}g_{kn,n^{\prime }k^{\prime }}^{r}$.

Therefore, the downlink signal-to-interference-plus-noise ratio (SINR) of UE *k* associated with mmWave SBS cluster $C_{k}$ and associated with mmWave SBS *n* can be, respectively, denoted by
(6)$$SINR_{k}=\frac{\sum_{n\in C_{k}}p_{n,k}G_{n,k}h_{n,k}}{I_{k,intra}+I_{k,inter}+B_{a}N_{0}}$$
(7)$$SINR_{n,k}=\frac{P_{n,k}G_{n,k}h_{n,k}}{I_{\mathrm{int}ra}+I_{\mathrm{int}er}+B_{a}N_{0}}$$
where $h_{n,k}$ is the channel gain from mmWave SBS *n* to UE *k*, which includes pathloss and shadow fading, given by simulation parameter. $B_{a}$ is the access link mmWave bandwidth, $N_{0}$ represents the noise power density, and $BN_{0}$ is additive white Gaussian noise (AWGN).

Then, as mentioned above, according to the Shannon formula, the access link achievable rate of UE *k* associated with mmWave SBS cluster $C_{k}$ and associated with mmWave SBS *n* can be, respectively, written as
(8)$$\begin{array}{c} R_{k}=B_{a}\log_{2}\left(1+SINR_{k}\right) \end{array}$$
(9)$$\begin{array}{c} R_{n,k}=B_{a}\log_{2}\left(1+SINR_{n,k}\right) \end{array}$$

Additionally, the backhaul link achievable rate from MBS 0 to SBS *n* can be given by
(10)$$\begin{array}{c} R_{0,n}=B_{b}\log_{2}\left(1+\frac{p_{0,n}G_{0,n}h_{0,n}}{B_{b}N_{0}}\right) \end{array}$$
where $h_{0,n}$ is the channel gain from MBS 0 to SBS *n*, and $B_{b}$ is the backhaul link mmWave bandwidth.

### 3.3. Problem Formulation

With an objective function to maximize the system throughput of mmWave network, our joint user association and power allocation problem is formulated as follows:(11)$$\begin{array}{cc} P0: & \;\;\max_{X,P}\sum_{k\in K}R_{k}\\ s.t.\\ \; & C1:0\leq \sum_{k\in K}x_{k,n}\leq K_{\max},\forall n\in N\\ \; & C2:0\leq \sum_{n\in N}x_{k,n}\leq N_{\max},\forall k\in K\\ \; & C3:p_{n,k}\geq 0,\sum_{k\in K}p_{n,k}\leq p_{n}^{\max},\forall k\in K,n\in N\\ \; & C4:R_{k}\geq R_{k,\min},\forall k\in K\\ \; & C5:\sum_{k\in K}R_{n,k}\leq R_{0,n},\forall n\in N\\ \; & C6:x_{k,n}\in \left\{0,1\right\},\forall k\in K,n\in N \end{array}$$
where C1 ensures that each mmWave SBS n∈N serves maximally $K_{max}$ UEs. C2 expresses that each UE k∈K associates maximally $N_{max}$ mmWave SBSs simultaneously. C3 implies that transmit power of each mmWave SBS n∈N is non-negative and the total power is not higher than the maximum transmission power for each mmWave SBS n∈N. C4 indicates that the access link achievable rate is not lower than QoS requirement for each UE. C5 means the sum UE rates are not larger than backhaul capacity for each mmWave SBS.

Note that problem P0 is a nonconvex mixed-integer nonlinear programming optimization problem due to continuous variable $p_{k,n}$, binary variable $x_{k,n}$, and nonconvex transmission rate $R_{k}$. This problem cannot be solved directly by utilizing the traditional optimization method. Alternating optimization is one of the effective methods to solve this kind of intractable optimization problem [46,47,48]. Therefore, we decouple it into two independent subproblems: user association and power allocation subproblem, and further obtain the suboptimal solution by utilizing the matching game theory and SCA method iteratively in the following section.

## 4. Proposed Joint User Association and Power Allocation Optimization Scheme

To solve the original problem P0, a novel iterative joint user association and power allocation optimization scheme is proposed. First, given fixed power allocation coefficients, many-to-many matching game theory is applied to solve the user association subproblem. Then, according to the obtained user association results, power allocation is further optimized by using the SCA method, where Lagrangian dual decomposition is used to solve the convex power allocation subproblem after conversion. Finally, we propose an iterative joint user association and power allocation algorithm.

### 4.1. User Association Subproblem

We first consider the user association subproblem with fixed power allocation coefficients. The user association subproblem is rewritten as
(12)$$\begin{array}{cc} P1: & \;\;\max_{X}\sum_{k\in K}R_{k}\\ s.t.\\ \; & C1:0\leq \sum_{k\in K}x_{k,n}\leq K_{\max},\forall n\in N\\ \; & C2:0\leq \sum_{n\in N}x_{k,n}\leq N_{\max},\forall k\in K\\ \; & C6:x_{k,n}\in \left\{0,1\right\},\forall k\in K,n\in N \end{array}$$

The matching game theory is a promising approach in wireless networks [49], and it is adopted to solve problem P1. The main idea of the matching game is to construct a mapping relationship between UE and SBS. The many-to-many matching model for user association is considered as shown in Figure 2, which is elaborated next.

**Definition** **1.** 
*(Many-to-Many Matching): For two mutually distinct sets K of UEs and N of mmWave SBSs, the many-to-many matching μ can be defined as a function from set K∪N into the set of all subsets of K∪N, such that*


*(1)* 
*$\mu \left(n\right)\subseteq K$ and $\left|\mu \left(n\right)\right|\leq K_{max}$;*
*(2)* 
*$\mu \left(k\right)\subseteq N$ and $\left|\mu \left(k\right)\right|\leq N_{max}$;*
*(3)* 
*$n\in \mu \left(k\right)\Leftrightarrow k\in \mu \left(n\right)$;*



*where conditions (1) and (2) depict quota constraints, which are based on constraint (C1) and (C2) in problem P1. Condition (3) represents the conditions required for the successful establishment of matching relationships.*


In this matching game model, the utility functions of UEs and mmWave SBSs are defined as their achievable rate. The UE utility function under matching μ is given as
(13)$$\begin{array}{c} U_{k}\left(\mu \right)=B_{a}\log_{2}\left(1+SINR_{k}\right) \end{array}$$

Similarly, the mmWave SBS utility function is denoted by
(14)$$\begin{array}{c} U_{n}\left(\mu \right)=\sum_{k\in \mu \left(n\right)}B\log_{2}\left(1+SINR_{n,k}\right) \end{array}$$

The preference lists of mmWave SBSs and UEs are established as
(15)$$\begin{array}{c} P_{SBS}=\left[P_{SBS}\left(1\right),\ldots ,P_{SBS}\left(n\right),\ldots ,P_{SBS}\left(N\right)\right] \end{array}$$
(16)$$\begin{array}{c} P_{UE}=\left[P_{UE}\left(1\right),\ldots ,P_{UE}\left(k\right),\ldots ,P_{UE}\left(K\right)\right] \end{array}$$
where $P_{SBS}\left(n\right)$ is the preference list of mmWave SBS *n*, and $P_{UE}\left(k\right)$ is the preference list of UE *k*. The preference relation ${≻}_{k}$ of each UE is denoted by
(17)$$\begin{array}{c} U_{k}\left(\mu \right)>U_{k}\left(\mu^{\prime }\right)\Leftrightarrow \left(\mu ,n\right){≻}_{k}\left(\mu^{\prime },n^{\prime }\right) \end{array}$$
which means that UE *k* is more preferable to match SBS *n* in μ than SBS $n^{\prime }$ in $\mu^{\prime }$ because UE *k* achieves a higher transmission rate in SBS *n*.

Similarly, the preference relation ${≻}_{n}$ of each mmWave SBS is defined by
(18)$$\begin{array}{c} U_{n}\left(\mu \right)>U_{n}\left(\mu^{\prime }\right)\Leftrightarrow \left(\mu ,k\right){≻}_{n}\left(\mu^{\prime },k^{\prime }\right) \end{array}$$

**Remark** **1.** 
*The problem P1 is a matching problem with externalities. The UE’s achievable rate is affected not only by the interference from matched mmWave SBSs but also by the external interference of other mmWave SBSs. Meanwhile, the throughput of each SBS is dependent on the UEs of the service and other SBSs.*


**Definition** **2.** 
*(Swap-Matching): Given two distinct UEs k and $k^{\prime }$, the swap-matching operation $\mu_{kn}^{k^{\prime }n^{\prime }}$ is described as*

$$\begin{array}{c} \mu_{kn}^{k^{\prime }n^{\prime }}=\left\{\mu \setminus \left\{\left(k^{\prime },\mu \left(k^{\prime }\right)\right),\left(k,\mu \left(k\right)\right)\right\}\right\}\cup \left\{\left(k^{\prime },\left\{\mu \left(k^{\prime }\right)\setminus \left\{n^{\prime }\right\}\right\}\cup \left\{n\right\}\right),\left(k,\left\{\mu \left(k\right)\setminus \left\{n\right\}\right\}\cup \left\{n^{\prime }\right\}\right)\right\} \end{array}$$

*where mmWave SBS $n\in \mu \left(k\right)$ and mmWave SBS $n^{\prime }\in \mu \left(k^{\prime }\right)$, and n and $n^{\prime }$ are two distinct mmWave SBS.*


**Definition** **3.** 
*(Swap-Blocking Pair): Given two distinct UEs k and $k^{\prime }$, $\left(k,k^{\prime }\right)$ is a swap-blocking pair if and only if*
(1)
*$\;\;\;\;U_{x}\left(\mu_{kn}^{k^{\prime }n^{\prime }}\right)\geq U_{x}\left(\mu \right),\forall x\in \left\{k,k^{\prime },n,n^{\prime }\right\}$;*
(2)
*$\;\;\;\;U_{x}\left(\mu_{kn}^{k^{\prime }n^{\prime }}\right)>U_{x}\left(\mu \right),\exists x\in \left\{k,k^{\prime },n,n^{\prime }\right\}$;*


*where $U_{x}$ is the utility value of the swap-matching member x. It can be observed that the swap-matching operation is only permitted when the utility value of all members does not reduce and at least one member increases after the exchange is completed.*


**Definition** **4.** 
*If there is no swap-blocking pair, matching μ is a two-side exchange-stable matching.*


Based on the above analysis, the user association process is shown in Algorithm 1, which is divided into three main steps, i.e., Initialization stage, Matching process stage, and Swap-matching process stage. First, the Initialization stage initializes the preference list of unmatched mmWave SBSs and UEs as per descending order of channel gain. After that, the Matching process stage initializes the association between mmWave SBSs and UEs according to maximum channel gain based on constraints (C1) and (C2) in problem P1. Finally, the Swap-matching process stage achieves stable matching between the mmWave SBSs and UEs by updating the matching state and the number of swap-blocking pairs.

To evaluate the proposed Algorithm 1, we theoretically prove convergence and stability in the following.

**Theorem** **1.** 
*(Convergence): Algorithm 1 converges to the final matching $\mu^{*}$ with a finite iteration number.*


**Proof** **.** The utility function value of UEs and mmWave SBSs are increased after each swap according to Algorithm 1. Since the transmit power and spectrum resources are limited, the utility function value of UEs and mmWave SBSs has an upper bound. When the utility function value of UEs and mmWave SBSs is saturated, the swap operations finish. Therefore, Theorem 1 can be proved.   □

**Theorem** **2.** 
*(Stability): The final matching $\mu^{*}$ of Algorithm 1 achieves the two-sided stability.*


**Proof** **.** Based on *Definition 3*, if there is a swap-blocking pair $\left(k,k^{\prime }\right)$ in $\mu^{*}$, which satisfies $U_{x}\left(\mu_{kn}^{k^{\prime }n^{\prime }}\right)\geq U_{x}\left(\mu \right)),\forall x\in \left\{k,k^{\prime },n,n^{\prime }\right\}$ and $U_{x}\left(\mu_{kn}^{k^{\prime }n^{\prime }}\right)>U_{x}\left(\mu \right),\exists x\in \left\{k,k^{\prime },n,n^{\prime }\right\}$. Algorithm 1 does not complete until all swap-blocking pairs are eliminated. Therefore, matching $\mu^{*}$ is not the final matching of Algorithm 1, which conflicts with the original assumption. Theorem 2 can be proved.   □

**Algorithm 1** The MMUA Algorithm.1: **Initialization:** Initialize preference lists $P_{SBS}$ and $P_{UE}$ as per descending order2: of channel gain. Initialization SBS matching $\mu \left(n\right)=\emptyset$, UE matching $\mu \left(k\right)=\emptyset$3: and the number of swap-blocking pair $N_{swap}=1$.4:  **Matching process:**5:   **While** ($P_{UE}\neq \emptyset$ or $\exists k\in K:\left|\mu \left(k\right)\right|<N_{\max}$) **do**6:     **for** each k∈K **do**7:       UE *k* propose to match first mmWave SBS in $P_{UE}\left(k\right)$, clear this index from8:      $P_{UE}\left(k\right)$.9:     **end for**10:    **for** each n∈N **do**11:       Accept the first $K_{max}$ maximum channel gain UEs, reject the rest of UEs into12:      unmatched UEs.13:    **end for**14:   **end while**15: **Swap-matching process:**16:   **While** $N_{swap}\neq 0$ **do**17:     **for** each k∈K, $n\in \mu \left(k\right)$ **do**18:      **for** each $k^{\prime }\in K\setminus k$, $n^{\prime }\in \mu \left(k^{\prime }\right)$ **do**19:        Calculate the utility value of all mmWave SBSs and UEs before and after20:       swapping, and determine whether $\left(k,k^{\prime }\right)$ is swap-blocking.21:       **if** *k* and $k^{\prime }$ form a swap-blocking pair **then**22:        Update matching state μ to $\mu_{kn}^{k^{\prime }n^{\prime }}$.23:        Update the number of swap-blocking pair $N_{swap}$.24:      **else**25:        Keep the original matching state.26:       **end if**27:      **end for**28:     **end for**29:    **end while**30: **Output**: A final matching $\mu^{*}$.

### 4.2. Power Allocation Subproblem

According to the obtained UE association matrix $X=X^{*}$ based on Algorithm 1, power allocation is further optimized in this subsection. The power allocation subproblem can be written as
(19)$$\begin{array}{cc} P2: & \;\;\max_{P}\sum_{k\in K}R_{k}\\ s.t.\\ \; & C3:p_{n,k}\geq 0,\sum_{k\in K}p_{n,k}\leq p_{n}^{\max},\forall k\in K,n\in N\\ \; & C4:R_{k}\geq R_{k,\min},\forall k\in K\\ \; & C5:\sum_{k\in K}R_{n,k}\leq R_{0,n},\forall n\in N \end{array}$$

Note that the nonconvexity of objective function (19) and constraints (C4) and (C5) determines that problem P2 is nonconvex. To solve it, the SCA method is adopted to convert the nonconvex problem P2 to convex subproblems. Subsequently, we apply concave lower bound and convex upper bound approximations to achievable rates.

**Proposition 1.** 
*The concave lower bound $L_{k}$ of the access link achievable rate of UE k associated with mmWave SBS cluster $C_{k}$ is*

(20)
$$\begin{array}{c} L_{k}\overset{\Delta }{\;=}B_{a}\log_{2}\left(N+f_{k}\left(P\right)\right)-B_{a}\log_{2}\left(N+g_{k}\left(P^{\tau }\right)\right)+\frac{B_{a}\nabla g_{k}\left(P^{\tau }\right)\left(P-P^{\tau }\right)}{\ln2\left(N+g_{k}\left(P^{\tau }\right)\right)} \end{array}$$


*and the convex upper bound $U_{n,k}$ of the access link achievable rate of UE k associated with mmWave SBS n is*

(21)
$$\begin{array}{c} U_{n,k}\overset{\Delta }{\;=}B_{a}\log_{2}\left(N+f_{n,k}\left(P^{\tau }\right)\right)+\frac{B_{a}\nabla f_{n,k}\left(P^{\tau }\right)\left(P-P^{\tau }\right)}{\ln2\left(N+f_{n,k}\left(P^{\tau }\right)\right)}-B_{a}\log_{2}\left(N+g_{k}\left(P\right)\right) \end{array}$$

*where
∇
denotes the gradient operation. $f_{k}\left(P\right)=\sum_{n\in C_{k}}p_{n,k}G_{n,k}h_{n,k}+I_{\mathrm{int}ra}+I_{\mathrm{int}er}$, $g_{k}\left(P\right)=I_{\mathrm{int}ra}+I_{\mathrm{int}er}$, and $f_{n,k}\left(P\right)=p_{n,k}G_{n,k}h_{n,k}+I_{\mathrm{int}ra}+I_{\mathrm{int}er}$.*


**Proof.** Please see Appendix A.    □

Then, by substituting upper and lower bounds $L_{k}$ and $U_{n,k}$ into the optimization problem P2, the problem P2 is rewritten as
(22)$$\begin{array}{cc} P3: & \;\;\max_{P}\sum_{k\in K}L_{k}\\ s.t.\\ \; & C3:p_{n,k}\geq 0,\sum_{k\in K}p_{n,k}\leq p_{n}^{\max},\forall k\in K,n\in N\\ \; & C^{\prime }4:L_{k}\geq R_{k,\min},\forall k\in K\\ \; & C^{\prime }5:\sum_{k\in K}U_{n,k}\leq R_{0,n},\forall n\in N \end{array}$$

It can be proved that problem P3 is convex. Then, a low-complexity Lagrangian dual decomposition algorithm in [50] is adopted to obtain its optimal solution. Problem P3 is solved by dealing with its dual problem. The Lagrangian function of problem P3 is written as
(23)$$\begin{array}{c} L\left(P,\lambda ,\mu ,\upsilon \right)=\sum_{k\in K}L_{k}+\sum_{n\in N}\left(\lambda_{n}\left(p_{n}^{\max}-\sum_{k\in K}p_{n,k}\right)\right)\\ +\sum_{k\in K}\mu_{k}\left(L_{k}-R_{k,\min}\right)+\sum_{n\in N}\upsilon_{n}\left(R_{0,n}-\sum_{k\in K}U_{n,k}\right) \end{array}$$
where λ, μ and υ are the dual variables. The dual problem of problem P3 is formulated as
(24)$$\begin{array}{c} P4:\min_{\lambda ,\mu ,\upsilon }d\left(P,\lambda ,\mu ,\upsilon \right)\\ s.t.\lambda \geq 0,\mu \geq 0,\upsilon \geq 0 \end{array}$$
and its Lagrange dual function can be given as
(25)$$\begin{array}{c} d\left(\lambda ,\mu ,\upsilon \right)=\max_{P}L\left(P,\lambda ,\mu ,\upsilon \right) \end{array}$$

**Proposition 2.** 
*The optimal solution of dual problem P4 can be obtained by updating Equations (26)–(29) until the iteration converges.*

(26)
$$\begin{array}{c} p_{n,k}\left(s+1\right)={\left[p_{n,k}\left(s\right)-\delta_{p_{n,k}}\frac{\partial L}{\partial p_{n,k}}\right]}^{+} \end{array}$$


(27)
$$\lambda_{n}\left(s+1\right)={\left[\lambda_{n}\left(s\right)-\delta_{\lambda_{n}}\left(p_{n}^{\max}-\sum_{k\in K}p_{n,k}\right)\right]}^{+}$$


(28)
$$\mu_{k}\left(s+1\right)={\left[\mu_{k}\left(s\right)-\delta_{\mu_{k}}\left(L_{k}-R_{k,\min}\right)\right]}^{+}$$


(29)
$$\upsilon_{n}\left(s+1\right)={\left[\upsilon_{n}\left(s\right)-\delta_{\upsilon_{n}}\left(R_{0,n}-\sum_{k\in K}U_{n,k}\right)\right]}^{+}$$


*where index s represent the number of iterations, and $\delta_{\lambda_{n}}$, $\delta_{\mu_{k}}$, and $\delta_{\upsilon_{n}}$ denote sufficiently small step sizes, ${\left[x\right]}^{+}=\max\left\{0,x\right\}$. $\frac{\partial L}{\partial p_{n,k}}$ is the first partial derivative, which is given by*

(30)
$$\begin{array}{c} \frac{\partial L}{\partial p_{n,k}}=\frac{B_{a}\left(1+\mu_{k}\right)}{\ln2}\left(\frac{1}{N+f_{k}\left(P\right)}\frac{\partial f_{k}\left(P\right)}{\partial p_{n,k}}-\frac{1}{N+g_{k}\left(P^{\tau }\right)}\frac{\partial g_{k}\left(P\right)}{\partial p_{n,k}}\right)\\ -\lambda_{n}-\frac{B_{a}\upsilon_{n}}{\ln2}\left(\frac{1}{N+f_{n,k}\left(P^{\tau }\right)}\frac{\partial f_{n,k}\left(P^{\tau }\right)}{\partial p_{n,k}}-\frac{1}{N+g_{k}\left(P\right)}\frac{\partial g_{k}\left(P\right)}{\partial p_{n,k}}\right) \end{array}$$

*where*

(31)
$$\begin{array}{c} \frac{\partial \left(g_{k}\left(P\right)\right)}{\partial p_{n,k}}=\left\{\begin{array}{cc} {\bar{G}}_{nk^{\prime },kn}h_{n,k^{\prime }}, & \mathrm{if}\;\;k\neq k^{\prime }\\ 0,\;\;\;\;\;\;\;\;\;\;\;\;\;\; & \mathrm{if}\;\;k=k^{\prime } \end{array}\right. \end{array}$$


(32)
$$\begin{array}{c} \frac{\partial \left(f_{n,k}\left(P\right)\right)}{\partial p_{n,k}}=\left\{\begin{array}{cc} {\bar{G}}_{nk^{\prime },kn}h_{n,k^{\prime }}, & \mathrm{if}\;\;k\neq k^{\prime }\\ G_{n,k}h_{n,k},\;\;\;\; & \mathrm{if}\;\;k=k^{\prime } \end{array}\right. \end{array}$$


(33)
$$\begin{array}{c} \frac{\partial \left(f_{k}\left(P\right)\right)}{\partial p_{n,k}}=\left\{\begin{array}{cc} {\bar{G}}_{nk^{\prime },kn}h_{n,k^{\prime }}, & \mathrm{if}\;\;k\neq k^{\prime }\\ G_{n,k}h_{n,k},\;\;\;\; & \mathrm{if}\;\;k=k^{\prime } \end{array}\right. \end{array}$$



**Proof.** Please see Appendix B.    □

The detailed procedure of the proposed SCA-based power allocation algorithm is shown in Algorithm 2. Note that according to [50,51], Algorithm 2 converges to a Karush–Kuhn–Tucker (KKT) solution. The objective function value of each iteration is not lower than the value in the previous iteration, which indicates that system throughput does not decrease as the iteration number increases. In addition, the QoS requirements of UEs, backhaul capacity, and transmit power of mmWave SBSs limit the increase in the system throughput, which makes Algorithm 2 converge.

**Algorithm 2** SCA-based Power Allocation Algorithm1: **Initialization:** Lagrange multipliers $\lambda =zeros\left(N\right)$, $\mu =zeros\left(K\right)$, $\upsilon =zeros\left(N\right)$,2: power allocation pn,k0=pnmaxpnmaxKmaxKmax, tolerance ε, maximum number of iterations3: $\tau_{max}$, and iteration number τ=0.4: **repeat**5:   **repeat**6:     With $L_{k}\left(\tau \right)$ and $U_{n,k}\left(\tau \right)$ fixed, calculate $p_{n,k}$ according to (26);7:     Update $\lambda_{n}$ according to (27);8:     Update $\mu_{k}$ according to (28);9:     Update $\upsilon_{n}$ according to (29);10:  **until** λ, μ, υ are convergent.11:  Update $L_{k}\left(\tau \right)$ and $U_{n,k}\left(\tau \right)$.12:  Update τ=τ+1.13: **until** $\tau =\tau_{max}$ or convergent.14: **Output**:The optimal transmit power $p_{n,k}^{*}$, and achievable rate ${L_{k}}^{*}$, ${U_{n,k}}^{*}$.

### 4.3. Joint User Association and Power Allocation Optimization Algorithm

After the solution of each subproblem is obtained, we give an overall algorithm for solving problem P0. We propose the iterative joint user association and power allocation (JUAPA) algorithm. The detailed procedure of the proposed JUAPA algorithm is presented in Algorithm 3.

First, we initialize the power allocation, which is expressed as
(34)pn,k0=pnmaxpnmaxKmaxKmax,∀n∈N,k∈K

**Algorithm 3** The JUAPA Algorithm1: **Input:** MmWave SBS quota $N_{max}$, UE quota $K_{max}$, QoS requirement $R_{k,min}$, and2: limited backhaul capacity $R_{0,n}$.3: **Initialization:** Initialize $p_{n,k}^{0}$ based on (34), feasible $X^{0}$ and iteration number t=0.4: **Repeat**5:   **Step 1: User association**6:     Update preference list $P_{UE}$ and $P_{SBS}$ based on (13) and (14).7:     Update user association $X^{t}$ with fixed $P^{t}$ by Algorithm 1.8:   **Step 2: Power allocation**9:     Update power allocation $P^{t}$ with fixed $X^{t}$ by Algorithm 2.10:     Update t=t+1.11: **Until** $X^{t}=X^{t-1}$.12: **Output**: The stable result $X^{t}$ and $P^{t}$.

The feasible result of user association $X^{0}$ is obtained based on Algorithm 1. Then, each iteration of the JUAPA algorithm consists of two steps: user association and power allocation. In step 1, the user association subproblem P1 is solved with fixed $P^{t}$. With the newly updated preference list $P_{UE}$ and $P_{SBS}$ based on (13) and (14), the user association matrix $X^{t}$ can be obtained by Algorithm 1. In step 2, based on $X^{t}$ power allocation, the subproblem P2 is solved with fixed $X^{t}$, where power allocation matrix $P^{t}$ can be updated by Algorithm 2. Furthermore, as the number of iterations increases, there is an upper bound on the monotonic increase in objective value, so the entire iterative JUAPA algorithm converges [7,11].

### 4.4. Complexity Analysis

In this subsection, we evaluate the computational complexity of the JUAPA algorithm. Note that in Algorithm 1, the number of matching operations is calculated as $O\left(NK^{2}\right)$, and the number of swap operations is $O\left(NK\right)$. Therefore, the complexity of Algorithm 1 is $O\left(NK^{2}+NK\right)$. The complexity of exhaustive searching (ES) algorithms increases exponentially with the number of UEs and mmWave SBSs. Algorithm 1 greatly reduces complexity compared with an ES algorithm. In Algorithm 2, the complexity of each iteration of power allocation optimization is $O\left(N^{2}K^{2}\right)$. Therefore, the complexity of Algorithm 1 is $O\left(T_{pa}\left(N^{2}K^{2}\right)\right)$, where $T_{pa}$ is the iteration number of Algorithm 2. Therefore, the total complexity of our proposed JUAPA algorithm (Algorithm 3) is mainly related to user association phase and power allocation optimization phase, which can be given as: $O\left(T_{oa}\left(NK^{2}+NK+T_{pa}\left(N^{2}K^{2}\right)\right)\right)$, where $T_{oa}$ is the iteration number of Algorithm 3.

## 5. Numerical Results and Discussion

### 5.1. Simulation Parameters

The different simulation parameters are considered as follows. Assume mmWave SBSs are randomly located according to the homogeneous Poisson point process (PPP) model with density $\lambda_{N}$, and UEs are uniformly located with density $\lambda_{K}$. Let $\lambda_{N}$ = 100/km${}^{2}$, $\lambda_{K}$ = 200/km${}^{2}$ by default. We model pathloss by $PL=32.4+21\log_{10}\left(d\right)+20\log_{10}\left(f_{c}\right)$, where *d* is the distance between UE and mmWave SBS, given in (km), and $f_{c}$ is center frequency, given in (MHz). Reyleigh fading is used to model the channel. The other key simulation parameters are listed in Table 1.

### 5.2. Simulation Results and Analysis

To verify the effectiveness of our algorithm, we compare the performance of the JUAPA algorithm with ES, best channel gain, max-SINR [52], random matching, and min-distance algorithms. We use these algorithms to optimize the user association, and other procedures are the same as the JUAPA algorithm. Take the ES algorithm as an example; the ES algorithm enumerates all user association results and executes Algorithm 2 to calculate system throughput. Subsequently, we choose the result with the highest system throughput among the iterative ES algorithms as our optimization solution.

To evaluate the convergence of our algorithm, the performance of the JUAPA algorithm is compared with that of the ES algorithm. Figure 3 illustrates the system throughput (i.e., the sum rate of UEs in the network) comparison between the JUAPA algorithm and ES algorithm, where UE density is 200/km${}^{2}$ and mmWave SBS density is 100/km${}^{2}$. It is observed that the proposed JUAPA algorithm converges in a finite iteration number. Furthermore, the performance of the JUAPA algorithm gradually approaches that of the ES algorithm as the iteration number increases. It is also observed that there is still a gap between the JUAPA algorithm and ES algorithm. The reason is that the original problem is a nonconvex optimization problem, and the JUAPA algorithm is unable to search globally in the feasible region, it can only obtain suboptimal solutions.

Figure 4 and Figure 5 demonstrate the convergence of Algorithm 1 and Algorithm 2, respectively, where UE density is 200/km${}^{2}$ and mmWave SBS density is 100/km${}^{2}$. The algorithm converges when the increment of the system throughput is no greater than error $10^{-4}$. It is seen that both of them converge after a finite iteration number, which verifies the correctness of Theorems 1 and 2.

Next, we show the impact of UE and mmWave SBS density on the network performance. Figure 6 depicts the average UE rate versus the densities of UEs. It is observed that with the increase in UE density, the average UE rate becomes decreased. One reason is that the limited transmit power of mmWave SBSs will be shared by more UEs, which will reduce the power received by each UE, resulting in a lower average UE rate. Another reason is the high UE density increases the number of LoS paths between interfering mmWave SBSs and UEs, which can make the interference be more severe. Although the high UE density decreases the average UE rate, the performance of the JUAPA algorithm is higher than that of all other algorithms. It is also seen that compared with the max-SINR algorithm, the JUAPA algorithm has 20.51% performance gain when the density is 50/km${}^{2}$, and it has 33.93% performance gain when the density is 400/km${}^{2}$.

In Figure 7, the impact of the densities of mmWave SBSs on system throughput is illustrated. It is seen that higher mmWave density makes the network have higher system throughput. This is because there are more ideal mmWave SBSs for UE association, which effectively restrains beam interference and has higher CoMP diversity gain. In other words, compared with other algorithms, the proposed JUAPA algorithm enables each UE to associate suitable mmWave SBS clustering. Each UE is served by a desirable mmWave SBS cluster and obtains a higher achievable rate. It is also observed that the system throughput of the JUAPA algorithm is higher than that of other algorithms, except the ES algorithm. Compared with the max-SINR algorithm, the proposed JUAPA algorithm improves system throughput by around 49.8% when the density is 25 /km${}^{2}$ and 18.19% when the density is 200/km${}^{2}$.

Figure 8 shows the relation between system throughput and QoS requirement value of UEs. With the increase in QoS value, system throughput becomes decreased. The reason is that higher QoS requirement value will result in unsuccessful transmission. Compared with the max-SINR algorithm, the proposed JUAPA algorithm has 22.89% performance gain when QoS requirement value is 50 Mbps and has 55.51% performance gain when QoS requirement value is 400 Mbps. This shows that the JUAPA algorithm has achieved higher reliability and system throughput.

Figure 9 demonstrates the relation between system throughput and maximum number of associated mmWave SBSs $N_{max}$ using different algorithms. As $N_{max}$ increases, the CoMP diversity gain of each UE also increases. Compared with the max-SINR algorithm, the proposed JUAPA algorithm has 34.21% performance gain when $N_{max}=1$ and has 22.76% performance gain when $N_{max}=6$.

It can be observed that the performance of the JUAPA algorithm in this paper is 95% above that of the ES algorithm for system throughput, average UE rate, and QoS satisfaction, effectively validating the superiority of the JUAPA algorithm.

## 6. Conclusions

This paper investigates the system throughput maximization problem for CoMP transmission in a downlink ultra-dense mmWave network by jointly considering user association and power allocation. This optimization problem is decoupled into two subproblems: user association and power allocation, which are solved by alternating optimization methods. First, we design a novel MMUA algorithm with externalities. Afterwards, a low-complexity SCA algorithm is designed to solve power allocation, where Lagrangian dual decomposition is adopted to solve the converted convex power allocation problem. Finally, the results confirm that the performance of the proposed JUAPA algorithm are close to those of the ES algorithm, which greatly reduces complexity. Compared with traditional algorithms, the JUAPA algorithm has significant advantages in improving system throughput and satisfying QoS requirements of UEs.

## Figures and Tables

**Figure 1 entropy-25-00409-f001:**
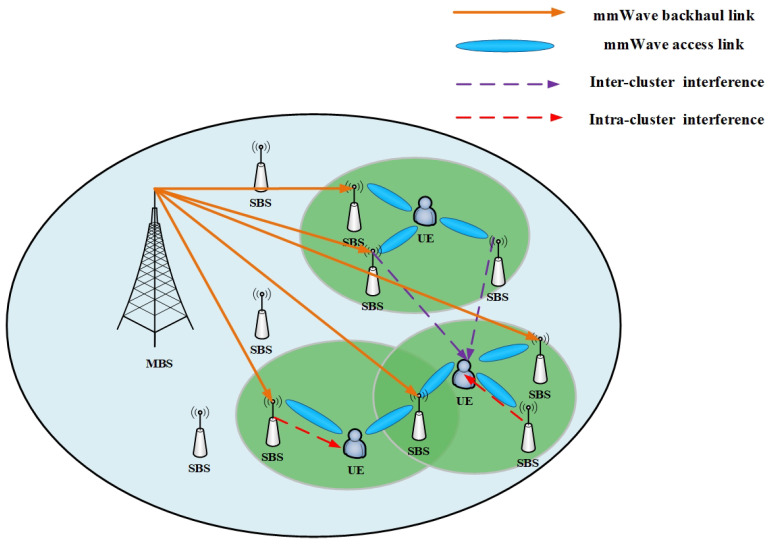
Network architecture.

**Figure 2 entropy-25-00409-f002:**
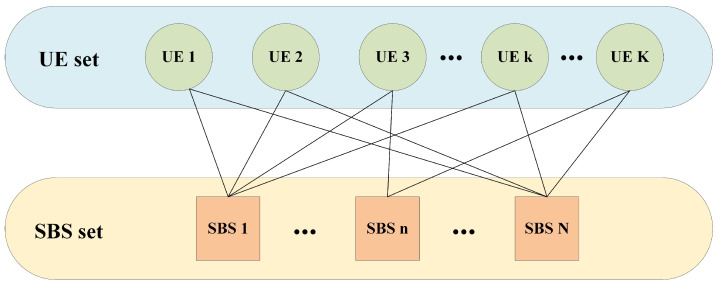
Many-to-many matching model for user association.

**Figure 3 entropy-25-00409-f003:**
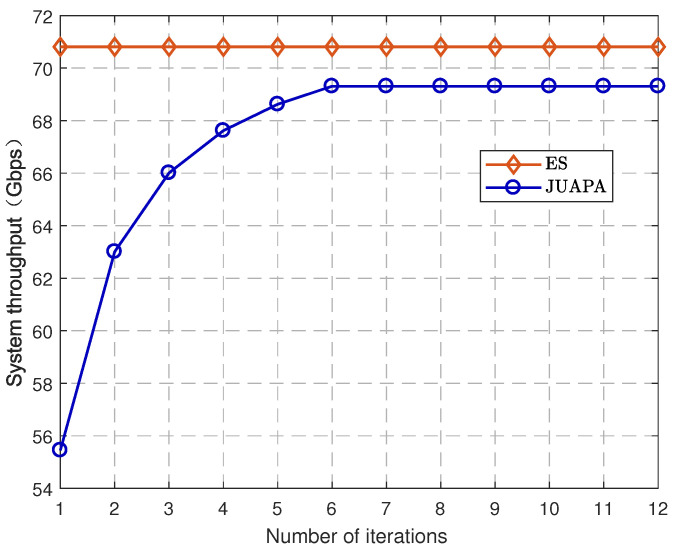
Convergence of JUAPA algorithm.

**Figure 4 entropy-25-00409-f004:**
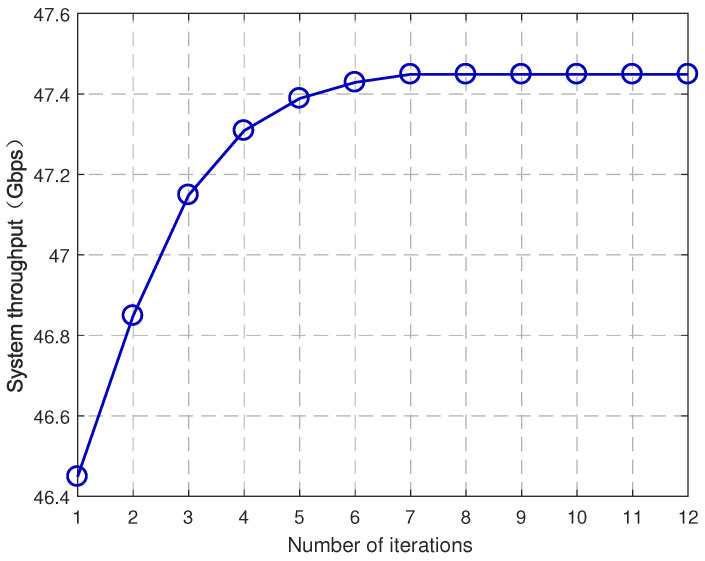
Convergence of user association algorithm.

**Figure 5 entropy-25-00409-f005:**
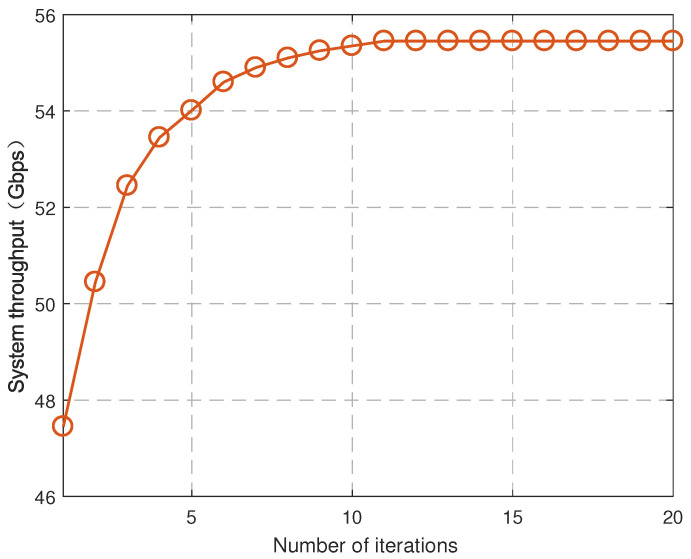
Convergence of power allocation algorithm.

**Figure 6 entropy-25-00409-f006:**
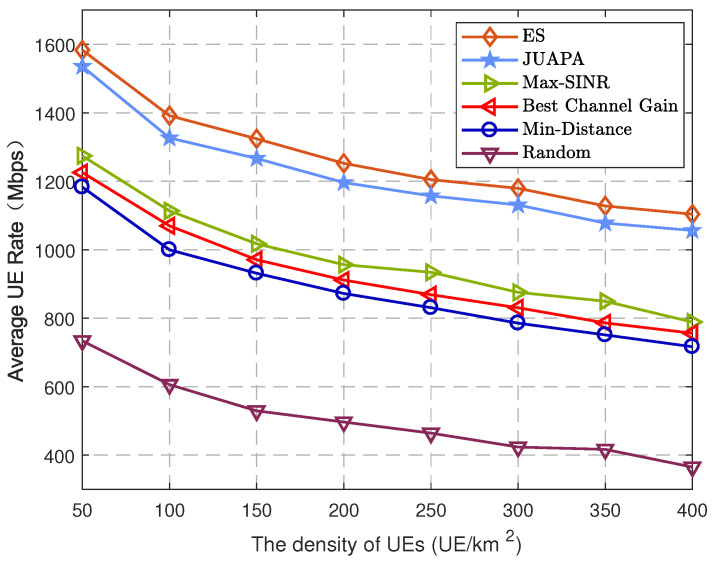
Average UE rate versus the density of UEs.

**Figure 7 entropy-25-00409-f007:**
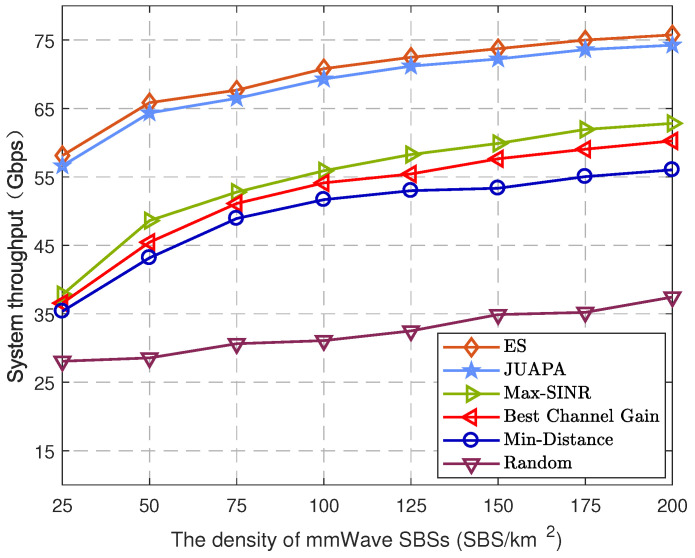
System throughput versus the density of mmWave SBSs.

**Figure 8 entropy-25-00409-f008:**
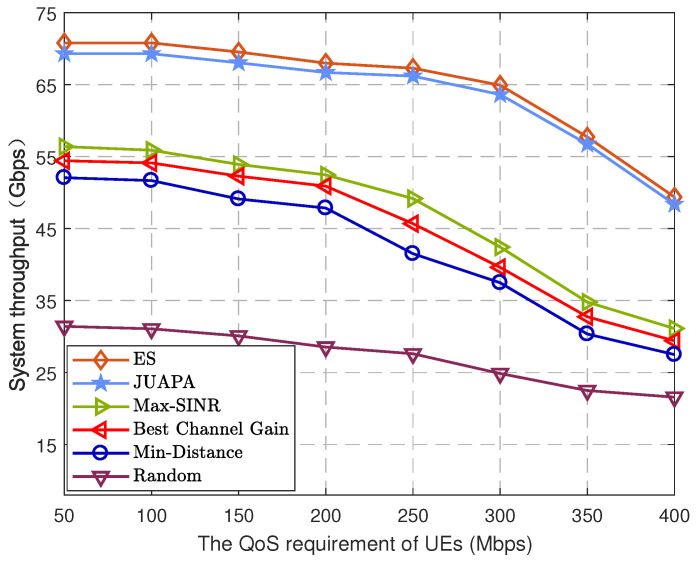
System throughput versus the QoS requirements of UEs.

**Figure 9 entropy-25-00409-f009:**
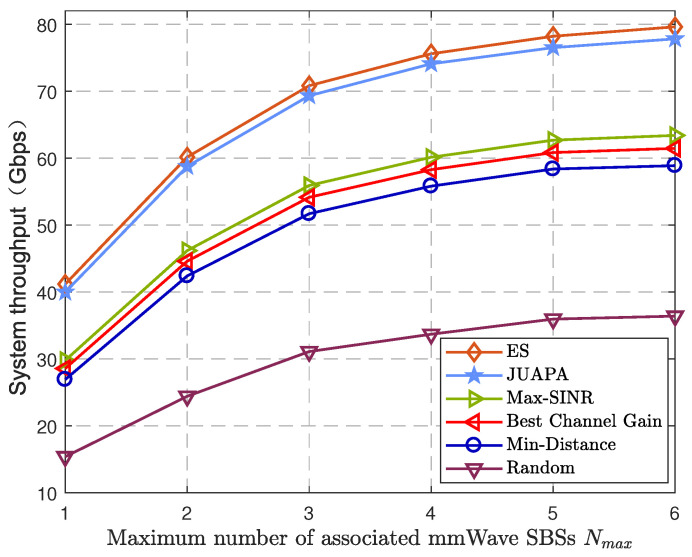
System throughput versus maximum number of associated mmWave SBSs $N_{max}$.

**Table 1 entropy-25-00409-t001:** Simulation parameters.

Parameters	Value
Bandwidth, $B_{a}$, $B_{b}$	0.2 GHz, 1.8 GHz
Coverage radius	300 m
Carrier frequency, $f_{c}$	28 GHz
Transmit power, $p_{0}^{\max}$, $p_{n}^{\max}$	50 dBm, 40 dBm
Noise spectral density, $N_{0}$	−174 dBm/Hz
Shadow fading standard deviation	10 dB
Beamwidth of UE *k*, $\theta_{k}^{r}$	$10^{\circ }$
Beamwidth of mmWave SBS *n*, $\theta_{n}^{t}$	$10^{\circ }$
Sidelobe gain, ε	0.1
LoS range constant, ρ	150 m
Maximum number of service UEs, $K_{\max}$	30
Maximum number of associated mmWave SBSs, $N_{\max}$	3
SCA error tolerance, ε	$10^{-4}$
QoS requirement, $R_{k,\min}$	100 Mbps

## Data Availability

Not applicable.

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
