# Peer review of "Backhaul Capacity-Limited Joint User Association and Power Allocation Scheme in Ultra-Dense Millimeter-Wave Networks"

_entropy, 2023, doi:10.3390/e25030409_

Round 1
Reviewer 2 Report
The paper studies the problem of joint user association and power allocation in ultra-dense mmWave networks under cell-free architecture; and derives a novel iterative algorithm to do so.
The work is interesting and can be useful for researchers working in this area.
My main comments are as follows:
1) Theorem 1: what does 'limited iteration number' mean here ? Can the maximal number of iterations be upper-bounded by a quantity that only depends on the system parameters ?
2) Complexity of ALgorithms 2 and 3: more details are required for the analysis.
3) Introduction section: the introduction section is weak and should be improved. In particular the authors completely missed to mention an important line of work, which studied, from an information-theoretic and communication angles, the problems of backhaul limited communication over cell-free type networks, both in uplink and downlink settings.
On this aspect the authors should mention that there are, roughly, two streams of works: those based on structured codes and those based on random (quantization) codes.
Important related missing works based on structered codes include:
L. Zhou and W. Yu, “Uplink multicell processing with limited backhaul via per-base-station successive interference cancellation,” IEEE Journal on Sel. Areas in Comm., vol. 31, no. 10, pp. 1981–1993, Oct. 2013.
I.E Aguerri, A Zaidi, "Lossy compression for compute-and-forward in limited backhaul uplink multicell processing", IEEE Transactions on Communications 64 (12), 5227-5238Y. Zhou and W. Yu, “Optimized backhaul compression for uplink cloud radio access network,” IEEE J. Sel. Areas Commun., vol. 32, no. 6, pp. 1295–1307, Jun. 2014.
S.-N. Hong and G. Caire, “Compute-and-forward strategies for cooperative distributed antenna systems,” IEEE Trans. Inf. Theory, vol. 59, no. 9, pp. 5227–5243, Sep. 2013.
I. E. Aguerri and A. Zaidi, “Compute-remap-compress-and-forward for limited backhaul uplink multicell processing,” in Proc. IEEE Int. Conf. Commun. (ICC), May 2016, pp. 1–6
Y. Tan and X. Yuan, “Compute-compress-and-forward: Exploiting asymmetry of wireless relay networks,” IEEE Trans. Signal Process., vol. 64, no. 2, pp. 511–524, Jan. 2016
S. H. Park, O. Simeone, O. Sahin, and S. Shamai (Shitz), “Multihop backhaul compression for the uplink of cloud radio access networks,” IEEE Trans. Veh. Technol., vol. 65, no. 5, pp. 3185–3199, May 2015.
A. Sanderovich, O. Somekh, H. Poor, and S. Shamai (Shitz), “Uplink macro diversity of limited backhaul cellular network,” IEEE Trans. Inf. Theory, vol. 55, no. 8, pp. 3457–3478, Aug. 2009.
I. E. Aguerri and A. Zaidi, “Lossy compression for compute-and-forward in limited backhaul uplink multicell processing,” in Proc. IEEE Inf. Theory Workshop (ITW), Sep. 2016.
Important related works based on compression for backhaul reduction in downlink include:
- C.-Y. Wang and M. Wigger and A. Zaidi, "On achievability for downlink cloud radio access networks with base station cooperation:, IEEE Trans. on Inf. Theory, Vol. 64, No. 08, Aug. 2018, pp. 5726-5742.
S.-H. Park, O. Simeone, O. Sahin, and S. Shamai (Shitz), “Robust and efficient distributed compression for cloud radio access networks,” IEEE Trans. Veh. Technol., vol. 62, no. 2, pp. 692–703, Feb. 2013.
S.-H. Park, O. Simeone, O. Sahin, and S. Shamai (Shitz), “Joint decompression and decoding for cloud radio access networks,” IEEE Signal Process. Lett., vol. 20, no. 5, pp. 503–506, May 2013.
Round 2
Reviewer 1 Report
No further remarks or comments.
Plese check units in the equation used to model path loss.
Reviewer 2 Report
The authors did not improve the introduction section and bibliography, by mentioning (and discussing) the various lines of work for backhaul-compression, for both uplink and downlink settings. Although the paper only deals with downlink, also discussing (in the introduction section, with few lines and references) the most relevant approaches is useful for the readers. The reason is that there exists a known duality between uplink and downlink. So, powerful techniques for uplink have their duals for downlink that can be bealt easily.
The authors need to improve this part. There is a huge literature on this; and this reviewer has already mentioned examples of relevant works (for both structured coding based techniques and those based on compression). For this reason, I feel that this paper needs still to go through another revision.
Round 3
Reviewer 2 Report
The authors have revised their manuscript in a satisfactory manner; I recommend Accept.